# In Silico Analysis of Functionalized Hydrocarbon Production Using Ehrlich Pathway and Fatty Acid Derivatives in an Endophytic Fungus

**DOI:** 10.3390/jof7060435

**Published:** 2021-05-29

**Authors:** Kristopher A. Hunt, Natasha D. Mallette, Brent M. Peyton, Ross P. Carlson

**Affiliations:** 1Center for Biofilm Engineering, Montana State University, Bozeman, MT 59717, USA; hunt0362@uw.edu (K.A.H.); natashamallette@gmail.com (N.D.M.); bpeyton@montana.edu (B.M.P.); 2Department of Chemical and Biological Engineering, Montana State University, Bozeman, MT 59717, USA

**Keywords:** endophyte, consolidated bioprocessing, efma, fba, cytosolic pyruvate dehydrogenase

## Abstract

Functionalized hydrocarbons have various ecological and industrial uses, from signaling molecules and antifungal/antibacterial agents to fuels and specialty chemicals. The potential to produce functionalized hydrocarbons using the cellulolytic, endophytic fungus, *Ascocoryne sarcoides*, was quantified using genome-enabled, stoichiometric modeling. In silico analysis identified available routes to produce these hydrocarbons, including both anabolic- and catabolic-associated strategies, and determined correlations between the type and size of the hydrocarbons and culturing conditions. The analysis quantified the limits of the wild-type metabolic network to produce functionalized hydrocarbons from cellulose-based substrates and identified metabolic engineering targets, including cellobiose phosphorylase (CP) and cytosolic pyruvate dehydrogenase complex (PDHcyt). CP and PDHcyt activity increased the theoretical production limits under anoxic conditions where less energy was extracted from the substrate. The incorporation of both engineering targets resulted in near-complete conservation of substrate electrons in functionalized hydrocarbons. The in silico framework was integrated with in vitro fungal batch growth experiments to support O_2_ limitation and functionalized hydrocarbon production predictions. The metabolic reconstruction of this endophytic filamentous fungus describes pathways for both specific and general production strategies of 161 functionalized hydrocarbons applicable to many eukaryotic hosts.

## 1. Introduction

*Ascocoryne sarcoides* is an endophytic fungus isolated from the *Eucryphia cordifolia* tree in Chilean Northern Patagonia and has been studied for its capacity to produce functionalized hydrocarbons (FHs) [1,2,3,4,5,6]. Endophytes are ubiquitous components of plant microbiomes that may contribute advantageous and essential functions to their plant host, such as tolerance to nutrient, water, temperature, and salinity stresses, as well as protective strategies against predation [7]. *A. sarcoides* produces a variety of FHs hypothesized to play a role in the symbiotic relationship with its host [8,9]. FHs produced by *A. sarcoides* also have uses as renewable chemicals, such as biofuels and flavor compounds, made from low-cost substrates, including cellulose and other agricultural wastes [10,11,12]. This consolidation of FH production from a diverse range of feedstocks makes *A. sarcoides* a candidate for bioprocessing [13]. The production of FHs by *A. sarcoides* has been studied in vitro at the organism level, but there have been minimal molecular-level pathway examinations and no in silico reconstructions of its metabolism [3].

Production of FHs has been shown to occur via four primary metabolic routes in eukaryotes: traditional fermentation, fatty acid production, linoleic acid degradation, and Ehrlich pathways. Traditional fermentation pathways produce low molecular weight byproducts from central metabolisms, such as ethanol and acetate. Fatty acid biosynthetic machinery can produce many FHs as a function of the initiating molecule, the number of elongation cycles, and endpoint of the final cycle. Examples include octane, octanol, and hexanoic acid [14]. Linoleic acid degradation can produce a variety of C_8_ to C_10_ FHs, such as 1-octen-3-ol, through the oxidation of double bonds and subsequent redox reactions [3]. Ehrlich pathways can produce various C_4_ to C_10_ FHs through the deamination and decarboxylation of amino acids or their derivatives, including 1-methyl-3-butanol [15]. Additionally, alcohols and acids from the described routes can be used to produce esters, further expanding the range of possible FHs [14,16,17]. A thorough, systems-based analysis of these metabolic strategies in an endophytic fungus has not been reported.

Acetyl-CoA is a building block for many FHs and their precursors, such as lipids and some amino acids [18]. The production and degradation of fatty acids in eukaryotes are functionally partitioned using acyl carrier protein (ACP) and coenzyme A (CoA). Production and degradation are also spatially partitioned through the localization of enzymes between the cytosol, mitochondria, and peroxisome [19]. Metabolite compartmentalization creates control points for balancing cellular energy generation in the mitochondria and fatty acid production in the cytosol. Acetyl-CoA can be transported between the mitochondria and the cytosol using three routes, with each having different cellular energy requirements (Figure 1) [20]. The ATP-citrate lyase (Acl) route cleaves citrate produced in the mitochondria into oxaloacetate and acetyl-CoA in the cytosol [21,22,23]. The acetate route converts acetyl-CoA to acetate in the mitochondria, transports the acetate to the cytosol, and reactivates it via acetyl-CoA synthetase (Acs). The acyl-carnitine route uses acyl-carnitine transport to move acyl groups between the cellular compartments for β-oxidation. Still, it cannot be the sole route for transporting acetyl-CoA from mitochondria to the cytosol without genetic engineering [18,23]. The pathways used to generate acetyl-CoA typically correlate with the metabolic strategy of the organism. Oleaginous yeasts generate most of their acetyl-CoA in the cytosol using the Acl enzyme; the citrate used to form acetyl-CoA originates from mitochondria and must be transported into the cytosol. Non-oleaginous yeasts generate most of their acetyl-CoA in the cytosol through pyruvate decarboxylase (Pdc), acetaldehyde oxidase, and Acs activities, a process that does not require organelle transport processes (Figure 1) [18].

Stoichiometric modeling is a computational systems biology approach that can identify ecologically relevant phenotypes and gene targets for optimizing metabolic network function. It does so by accounting for critical central metabolism properties, including enzyme compartmentalization and cellular energy costs for metabolite transport [24,25,26,27]. There are two primary forms of stoichiometric modeling; both require an initial representation of the metabolic network, commonly generated from genomic or other experimental data [28,29,30,31]. The metabolic network is represented as a set of linear equations that define genome enabled and abiotic metabolite conversions. Flux balance analysis (FBA) uses these linear equations, bounds on reaction rates, and an objective function (i.e., maximizing biomass production) to obtain an optimal flux distribution [32]. Elementary flux mode analysis (EFMA) uses the same set of linear equations to obtain the smallest set of genetically distinct flux distributions known as elementary flux modes (EFMs). The complete set of EFMs represents all possible steady-state metabolisms of the network through nonnegative linear combinations allowing for predictions of maximum yields and the effect of genetic modifications [28,33,34].

Fungal endophytes are an important natural component of plant microbiomes and potential consolidated bioprocess catalysts. The presented work is one of the first metabolic reconstructions of an endophytic fungus, with the first study examining production of therapeutic resveratrol [35], and maps both specific and general FH production strategies applicable to many eukaryotes. The results compare FH production from the carbon conserving Ehrlich pathway and electron concentrating fatty acid derivatives with respect to substrate carbon and electrons conservation, chain length, and functional groups. Additionally, the results quantify the stoichiometric limitations of FH production with respect to (i) the theoretically optimal chain length of products from cellulose, (ii) the theoretical limits of cellulosic feedstocks considering metabolic engineering targets, and (iii) the impact of culturing conditions on metabolite profiles. Comparison of theoretical limits with observed production trends provides the foundation for FH production with applicability to biotechnology and biochemical analysis of eukaryotic compartmentalization.

## 2. Materials and Methods

### 2.1. Construction of Fungal Endophyte Metabolic Model

An *A. sarcoides* NRRL 50072 metabolic model was constructed from the genome annotations and transcriptome data available at http://asco.gersteinlab.org (accessed on 30 April 2021) [3]; incomplete pathways were filled using assumed reactions based on completeness of a pathway and reference organisms (Appendix A). Enzyme localization was modeled based on predictions from TargetP 1.1 [36] and considered for two compartments, the mitochondria and cytosol. The macromolecular composition of biomass was modeled after *Thielavia terrestris* as a representative filamentous fungus [37]. Protein and RNA monomer composition was determined by the average distribution of all open reading frames in the genome and the ribosomal subunits, respectively. DNA monomer composition was modeled to represent the guanine-cytosine content of the genome. Micronutrients were assumed non-limiting and nitrogen supplied by ammonium, sulfur by sulfate, and phosphorous by phosphate following published *A. sarcoides* NRRL 50072 metabolite studies [1,2,3,4]. O_2_ was required to produce essential fatty acids and, therefore, biomass.

### 2.2. Stoichiometric Analyses

The stoichiometric model was compiled as described in the following section using Microsoft EXCEL and converted to appropriate formats using CellNetAnalyzer version 2014.1 [38,39]. Flux distributions were enumerated using either RegEFMTool version 2.0 [40] or the constraints-based reconstruction and analysis (COBRA) toolbox [41] run in MATLAB version 2016B. Gene regulatory rules excluding the simultaneous use of mutually exclusive reactions were applied to minimize the number of physiologically irrelevant possibilities (Appendix A). All computations were performed on a machine with a maximum of 2 Intel Xeon X5690s and 120 GB of memory.

## 3. Results

### 3.1. In Silico Reconstruction of Carbon Uptake, Compartmentalization, and Functionalized Hydrocarbon Production

The nutrient requirements to produce cellular energy and functionalized hydrocarbons (FHs) in *A. sarcoides* were influenced by oligosaccharide chain length and acetyl-CoA transport routes. *A. sarcoides* can utilize cellulose as a carbon substrate [5], which has been shown to be degraded to soluble oligosaccharides (i.e., 1–5 glucose units) extracellularly in other fungi [42]. The impact of oligosaccharide chain length on cellular energetics was quantified by modeling the representative carbon and electron sources glucose and cellopentaose. These substrates enter the cell through ABC transporters, the oligosaccharides are depolymerized independent of phosphate, and a hexose kinase activates monosaccharides. Production of cytosolic acetyl-CoA was modeled according to the annotated genes for the Acl and Acetate routes (Figure 1). A Pdc enzyme was not found in the *A. sarcoides* genome, and native acyl-carnitine transport is not expected to be a major route for acetyl-CoA transport from mitochondria to the cytosol [18,23].

Metabolic model reactions for FH production were built using genomic and metabolomic data. A consensus list of FHs was assembled by collating any FH species reported by at least three publications about *A. sarcoides* (Table 1). These FHs were modeled using traditional fermentation, fatty acid production, linoleic acid degradation, and Ehrlich pathways (Figure 2). Fatty acid pathways can produce a wide variety of FHs subject to the initiating molecule (e.g., acetyl-CoA or propanoyl-CoA), monomers incorporated during elongation (e.g., C_2_ and C_3_ monomers), and stage of the final reduction cycle. These variations in components and pathways have a combinatorial effect [14,43,44] and explain most FHs reported for *A. sarcoides* (Table 1). The metabolic model considered acetyl- and propanoyl-CoA as the initiators for alkyl chains, while only acetyl-CoA was considered an elongating monomer to avoid excessive computational complexity. This approach resulted in even- or odd-chained FHs depending on the initiator molecule, acetyl-CoA or propanoyl-CoA, and the stage of fatty acid elongation (Figure 2). Esterification of FHs from these pathways accounted for esters reported in vitro (Table 1). In total, 161 different excreted FHs composed of eight different functional groups ranging from C_3_ to C_20_ were modeled.

### 3.2. In Silico Analysis of Cellular Energy Production

Ecologically competitive phenotypes for cellular energy production were identified using resource allocation theory, postulating that the fittest cellular phenotypes minimize the amount of limiting resource required to make a product [45]. The theory was applied by calculating the carbon source and O_2_ required to produce cellular energy for each elementary flux mode (EFM) as defined by the Cmoles of carbon source and moles of O_2_, respectively, consumed to produce a mole of cellular energy (equal to one mole of ATP phosphodiester bonds). EFMs that minimized carbon and O_2_ required to produce cellular energy formed a trade-off surface (black line in Figure 3). EFMs that minimized the carbon required to produce cellular energy are to the left in the plot. The least carbon required to produce energy was achieved by metabolic strategies that completely oxidized cellulose to carbon dioxide and water and required 0.2 Cmoles of cellopentaose and 0.2 moles of O_2_ per mole of cellular energy. Under O_2_ limited conditions, phenotypes that secrete reduced carbon byproducts, such as acetate, acetaldehyde, and ultimately ethanol, became competitive (i.e., moving from the top left to the bottom right of the trade-off surface in Figure 3). Anoxic cellular energy production was predicted (1.3 Cmole cellopentaose per mole of cellular energy), but not anoxic growth due to an O_2_ requirement to produce sterols and other lipids (Appendix A). A Cmole of cellopentaose produced more cellular energy than a Cmole of glucose due to the modeled sugar transporters. Transport of one cellopentaose (i.e., five monomers of glucose) or one glucose into the cell required the same energy expenditure.

### 3.3. In Silico Analysis of Functionalized Hydrocarbon Production

FH production balances cellular energy production, electron trafficking, and resource utilization, which are all sensitive to nutrient limitations during cultivation. Trade-offs between carbon source and O_2_ requirements quantified competitive FH producing phenotypes. The carbon and O_2_ requirements to produce FHs were defined as the Cmoles of carbon source or moles of O_2_, respectively, consumed to make a Cmole of FH. Eight hydrocarbon functional groups (i.e., acids, esters, aldehydes, ketones, primary and secondary alcohols, alkanes, and alkenes) with chain lengths from C_3_ to C_20_ were modeled that represent most FHs reported for *A. sarcoides* (Table 2). FH producing metabolic strategies that required the least carbon source and O_2_ are to the left and bottom, respectively (Figure 4 and Figure 5). Metabolic strategies that produced a single, target FH of a specific chain length while minimizing carbon and O_2_ requirements are depicted by colored lines in Figure 4 and Figure 5. FHs generally required less carbon source to produce in the presence of O_2_ as quantified by the lowest Cmole of substrate per Cmole of FH (i.e., negative slopes of colored lines in Figure 4 and Figure 5). Production of Ehrlich pathway derivatives generally required fewer Cmoles of carbon source than fatty acid derivatives. The low molecular weight primary fermentation products formate, acetate, acetaldehyde, and ethanol were not considered as they were not the focus of this study.

#### 3.3.1. Ehrlich Pathway Derivatives

Resource requirements to produce Ehrlich pathway derivatives from 7 amino acids (i.e., methionine, valine, isoleucine, leucine, phenylalanine, tryptophan, and tyrosine) were governed under oxic conditions by the FH degree of reduction. Aromatic Ehrlich pathway derivatives, such as indole-3-acetate and 4-hydroxyphenylacetate, required the least carbon source of any considered FHs under oxic conditions (green lines in Figure 4), only propanoic acid required less under anoxic conditions (red line in Figure 4). These FHs retained a large fraction of the carbon from the carbon source and have moderate degrees of reduction that clustered around 4.5 (Figure 6). Conversely, Ehrlich pathway derivatives with higher degrees of reduction (i.e., 5–6 electron moles Cmole^−1^), such as isopentyl derivatives, retained less carbon source carbon (orange lines in panels for acids, aldehydes, and primary alcohols, yellow for esters of Figure 4). The trade-off between carbon retention and electron density was a function of electron conservation and substrate electron density, which are major design constraints for synthesizing a FH of interest.

Simulated production of Ehrlich pathway derivatives indicated an increase in Cmoles of substrate requirements as O_2_ availability decreased because more carbon source was oxidized for cellular energy production. Isopentyl derivatives were less sensitive to energy requirements than aromatic derivatives and required less carbon source to produce when O_2_ was limited. A similar trend was observed across examined functional groups (Figure 4). Aromatic FHs could not be produced under anoxic conditions because only the O_2_ requiring amine oxidase was found in the genome (i.e., a phenylpyruvate decarboxylase was not annotated in the *A. sarcoides* genome).

#### 3.3.2. Fatty Acid Derivatives

Efficient production of fatty acid derivatives was governed by their monomers and cellular energy requirements. Initiating fatty acids with propanoyl-CoA required less carbon source under oxic conditions than initiating it with acetyl-CoA (Figure 5). Propanoate was produced with high carbon source efficiency from 2-oxobutanoate, a precursor of isoleucine metabolism. However, as acetyl-CoA monomers were added to the propanoyl-CoA initiated molecule, the resulting FHs approached the theoretical limit of 1.5 Cmole of carbon source per Cmole of FH produced (Odd chain length acids in Figure 5). In contrast, acetyl-CoA initiated FHs were already at the theoretical limit independent of chain length (Even chain length acids in Figure 5). Fatty acid derived FHs all lost at least one carbon dioxide per monomeric acetyl-CoA but retained most if not all the electrons from the carbon source, as evidenced by the degrees of reduction approaching 6 (Figure 6). A similar trend was observed for all primary functional groups, whereas secondary functional groups were decarboxylated and lost the initial carbon savings associated with propanoyl-CoA. Fatty acid derivatives produced under low O_2_ availability required coproduction of other byproducts, such as ethanol and acetate, to produce the cellular energy required for their production and therefore required more carbon source (Figure 5). Acetyl-CoA-initiated fatty acids and aldehydes required formate coproduction under anoxic conditions to balance electrons. Therefore, as the functional group became more reduced (i.e., alcohols are more reduced than aldehydes which are more reduced than acids) and consumed additional electrons, the carbon source requirements decreased. Odd chain length fatty acids and aldehydes used propanoyl-CoA as the initiating molecule, which consumed the extra electrons and removed the requirement to coproduce formate.

FHs produced from linoleic acid degradation required more resources than those derived from fatty acids directly and could not be produced as the sole reduced byproduct. For example, all C_8_ derivatives were produced with a C_9_ or C_10_ coproduct and were nontrivial compared with other FHs. However, they required more resources than the C_18_ precursor from which they were derived.

### 3.4. In Silico Metabolic Engineering Targets

In silico models provide a framework for predicting network modifications to enhance catalytic function [20]. Two enzymatic modifications were tested using the presented *A. sarcoides* model for their capacity to decrease the cellular energy cost of metabolite transport: (1) the incorporation of a cellobiose phosphorylase (CP) and (2) the incorporation of cytoplasmic pyruvate dehydrogenase (PDH_cyt_).

#### 3.4.1. Cellobiose Phosphorylase

In silico analysis of recombinant CP activity demonstrated a reduced cellular energy requirement for sugar activation. CP conserved some of the energy in a β(1,4) glucose bond via sugar phosphorylation using inorganic phosphate. The energy conservation alleviated the cost of activating n-1 glucose molecules using ATP, where n is the number of monomers in the oligosaccharide being depolymerized. CP activity decreased the predicted requirements for carbon source and O_2_ for all FHs, except for 2-methyl-propanoate and 2-methyl-butanoate, which did not require a net input of energy to produce (Figure 7). This effect was most prominent under anoxic conditions where cellular energy production is more substrate intensive (Table 2). The Cmoles of substrate required to produce FHs decreased by as much as 17% when cellopentaose was the carbon source. No change was predicted for cultures grown with glucose (n = 1) as the monomeric carbon source does not have β(1,4) glucose bonds.

#### 3.4.2. Cytosolic Pyruvate Dehydrogenase

Transport of acetyl-CoA from the mitochondria to the cytosol constrains cytosolic production of fatty acids [23]. All acetyl-CoA transport routes described above require energy consumption to transfer mitochondria-derived acetyl-CoA to the cytosol, where it can be incorporated into fatty acids. PDH_cyt_ activity was hypothesized to uncouple cytosolic acetyl-CoA production from the energetic transport requirements and provide cytosolic reducing equivalents that increased the yield of fatty acid-derived FHs (dashed green line in Figure 1). This effect was most prominent in anoxic simulations, where the carbon source requirement for FH production decreased up to 30%. PDH_cyt_ activity had a less significant impact under oxic conditions (6% decrease in carbon source requirement, Table 2) due to the higher cellular energy yields per carbon source during O_2_ respiration (Figure 7). PDH_cyt_ activity did not impact Ehrlich pathway derivatives because they were produced in the mitochondria when derived from acetyl-CoA. The PDH_cyt_-based decreases in substrate requirements for FH production were similar for both cellopentaose and glucose.

#### 3.4.3. Concurrent Cellobiose Phosphorylase and Cytosolic Pyruvate Dehydrogenase Activity

The concurrent effect of both enzyme modifications decreased the requirement for carbon sources to produce FHs by up to 7 and 38% under oxic and anoxic conditions, respectively (Table 2). The largest reductions were observed for fatty acid derivatives as they are the most energy intensive and dependent on cytosolic acetyl-CoA. However, fatty acid derivatives were still constrained by a minimum theoretical limit of 1.5 Cmoles carbon source per Cmole of product due to the loss of carbon dioxide during the formation of acetyl-CoA from malonyl-CoA and due to electron conservation between carbon source and product. Exceptions to this are fatty acids derived from alternative initiating molecules, such as propanoyl-CoA.

## 4. Discussion

Functionalized hydrocarbons (FHs) produced by the endophytic fungi *A. sarcoides* have theoretical yields governed by the cellular energy, carbon, and electrons they require for production. The presented analysis quantified the theoretical yields of 161 FHs from cellulose-based carbon sources to understand their production for ecological or industrial purposes. The study identified the most efficient, theoretical production of the 161 different FHs based on criteria such as the conversion of a substrate, the number of Cmoles per product, and the presence of different secondary functional groups. The systems-based analysis identified FHs production properties that would not otherwise be obvious. Noteworthy FH production considerations in a eukaryotic host include: (1) Ehrlich pathway-derived FHs are generally more efficient at retaining substrate electrons and Cmoles than fatty acid-derived FHs. However, Ehrlich pathway derivatives were generally less reduced compounds. (2) Aromatic, Ehrlich pathway derived FHs required the least amount of carbon source under aerobic conditions of any FH considered in the study. (3) Contrary to the production of small primary fermentation products, including acetate, ethanol, and lactate, O_2_ limitation decreased the theoretical yield of FHs due to energy limitations. (4) FHs produced from isopentyl derivatives were less sensitive to O_2_ limitation than aromatic derivatives.

Fatty acid production can account for the most commonly reported FHs from *A. sarcoides* (Table 1). The presented stoichiometric model examined the effect of different precursors, including common central carbon metabolism (i.e., acetyl-CoA) and amino acid-derived intermediates (i.e., propanoyl-CoA). These two initiating molecules impacted the production of FHs through electron and energy balances. For example, the production of propanoyl-CoA required more electrons than acetyl-CoA, thereby serving as an effective electron sink, reducing the production of primary fermentation products like acetate and ultimately improving carbon retention during FH production.

Genetic modifications were examined in silico that impacted energy conservation and the transport and compartmentalization of acetyl-CoA. Modification of enzymatic activity (i.e., CP and PDH_cyt_) was predicted to reduce the carbon source requirements of FH production to the theoretical limits. The stoichiometric analysis predicted the addition of CP and PDH_cyt_ activities to FH producing cells could increase FH yields up to 38% on the carbon source (Table 2). CP activity reduced the activation cost of the carbon source, and PDH_cyt_ activity reduced the energetic requirement for cytosolic acetyl-CoA, increasing the yield of FHs. The combined impact of these modifications reduced the requirement for carbon sources to produce fatty acid derivatives to 1.5 Cmoles carbon source per Cmole of FH. These modifications and others, such as disrupting acetyl-CoA partitioning between the mitochondria and cytosol, have increased product yields and rates in other genetically modified organisms [22,46].

In vitro analysis of *A. sarcoides* NRRL 50072 cultures were performed under different O_2_ partial pressures to further interpret in silico results (Appendix A). The in vitro data supported predictions that cellular energy yields and, therefore, biomass yields would decrease with decreasing O_2_ availability (Appendix A). As the O_2_ partial pressure decreased, secretion of the primary fermentation products, acetate and ethanol, increased to balance cellular reducing equivalents and cellular energy production. In vitro changes in *A. sarcoides* growth and excreted products suggest that the transition to fermentative metabolism occurred when the headspace was less than ~10% O_2_ (Appendix A). The experimental yield of FHs produced per g of carbon source consumed did not change substantially with decreasing O_2_ partial pressures, but the diversity of secreted FHs increased (Appendix A). All in vivo FH yields were at least 10^4^-fold lower than theoretical limits, which indicated production was not for catabolic purposes but rather served other biological functions, like perhaps, communication molecules. Feedback mechanisms in metabolism and physiology, such O_2_ concentration and carbon source, may control the production of specific FHs in vivo [47,48]. Previous reports indicate that some plant tissues also have metabolic shifts at ~10% O_2_ partial pressure [49]. This may provide an in vivo feedback mechanism indicating an infected host, thereby triggering increased production of defensive compounds from its endophytic symbionts. For example, at ~21% O_2_ partial pressure, a common signaling molecule for hyphal growth, 1,3-octadiene, was observed. In contrast, more reduced biomass derivatives, such as Ehrlich pathway derivatives, were observed with decreased O_2_ partial pressure (Appendix A), suggesting a communication role linked to intracellular redox state.

## 5. Conclusions

Biological FH synthesis is metabolically costly, yet it is observed in many microorganisms spanning many ecosystems. The current study presents the most extensive stoichiometric analysis of FHs produced by either an endophytic or filamentous fungus. The results provide metabolic blueprints for producing 161 different FHs, quantifying the metabolic costs in terms of cellulose-based carbon source and O_2_ to produce each FH. The simulations also provide insights into natural conditions that would favor their production and provide targets for metabolic engineering of eukaryotic hosts to increase FH production efficiency.

## Figures and Tables

**Figure 1 jof-07-00435-f001:**
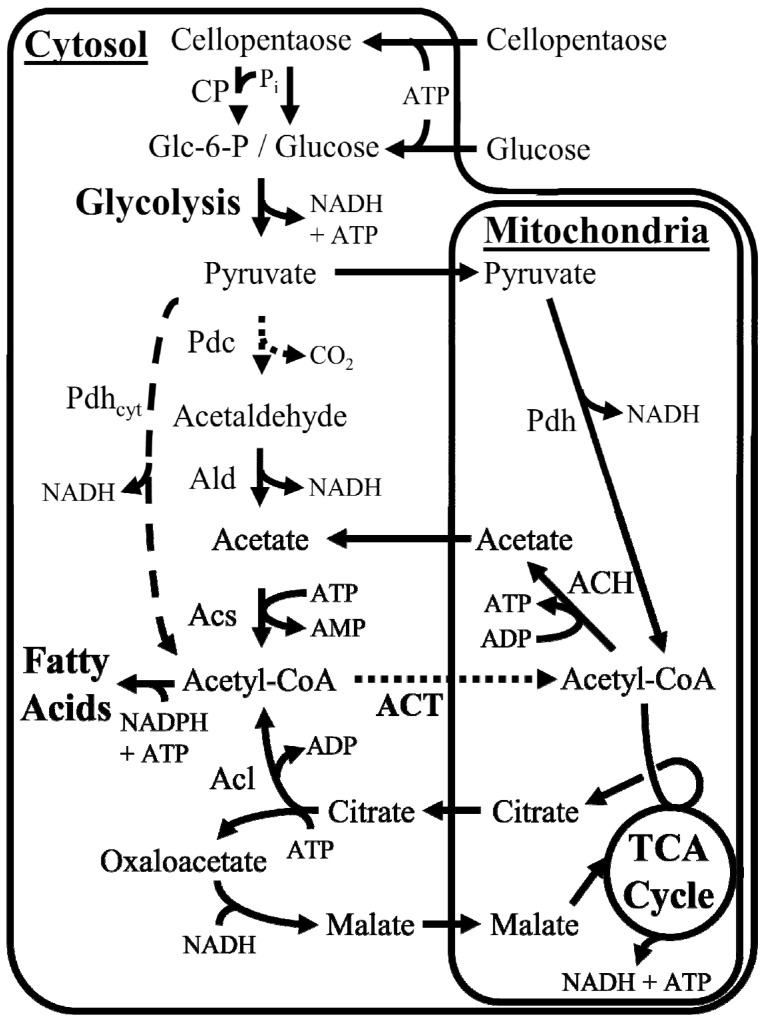
Overview of fungal acetyl-CoA metabolism. Common acetyl-CoA pathways in fungi are shown in solid arrows. A cytosolic pyruvate dehydrogenase (heavy dashed line) may increase functionalized hydrocarbon production but was not found in fungi and was not annotated in *Ascocoryne sarcoides* NRRL 50072. Enzyme abbreviations: ACH—acetyl-CoA hydrolase, Acl—ATP citrate lyase, Acs—acetyl-CoA synthetase, ACT—acylcarnitine transport, Ald—aldehyde dehydrogenase, CP—cellobiose phosphorylase, Pdh_cyt_—pyruvate dehydrogenase, Pdh_cyt_—cytosolic pyruvate dehydrogenase, Pdc—pyruvate decarboxylase.

**Figure 2 jof-07-00435-f002:**
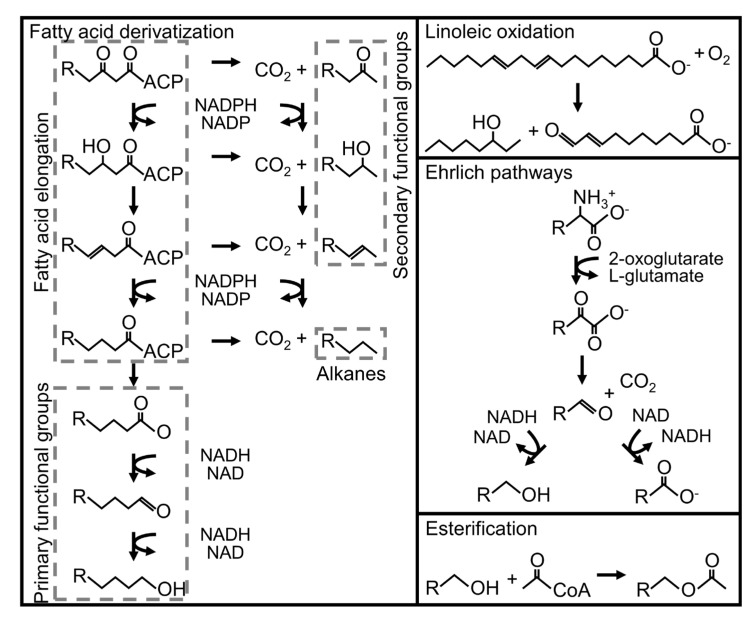
Overview of functionalized hydrocarbon producing pathways relevant to *A. sarcoides* NRRL 50072. Three pathways are depicted: fatty acid derivatization, linoleic oxidation, and Ehrlich pathways [3,14,15,43]. Products from these pathways and their esters account for most observed FHs. ACP refers to the acyl carrier protein associated with fatty acid synthesis, while R refers to carbon backbones. Associated genetic annotations can be found in the Appendix A.

**Figure 3 jof-07-00435-f003:**
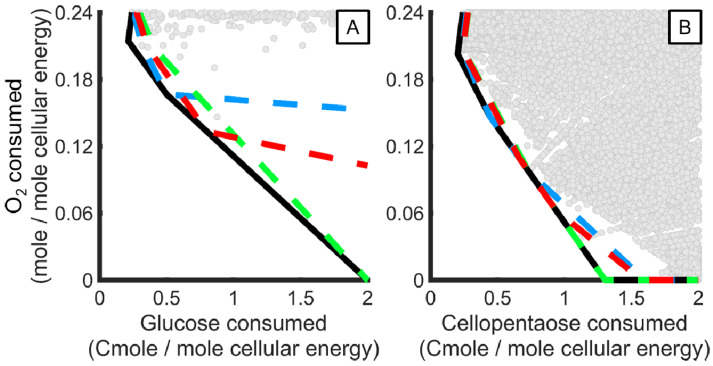
Metabolic strategies for cellular energy production from glucose (**A**) and cellopentaose (**B**) that minimize resource utilization span between complete oxidation and fermentation. Overall optimal O_2_ and carbon source utilization strategies for cellular energy production are highlighted with black boundaries. All possible metabolic strategies (elementary flux modes) are grey points. Metabolic strategies that produce only acetate (blue dashed line), acetaldehyde (red dashed line), or ethanol (green dashed line) are highlighted.

**Figure 4 jof-07-00435-f004:**
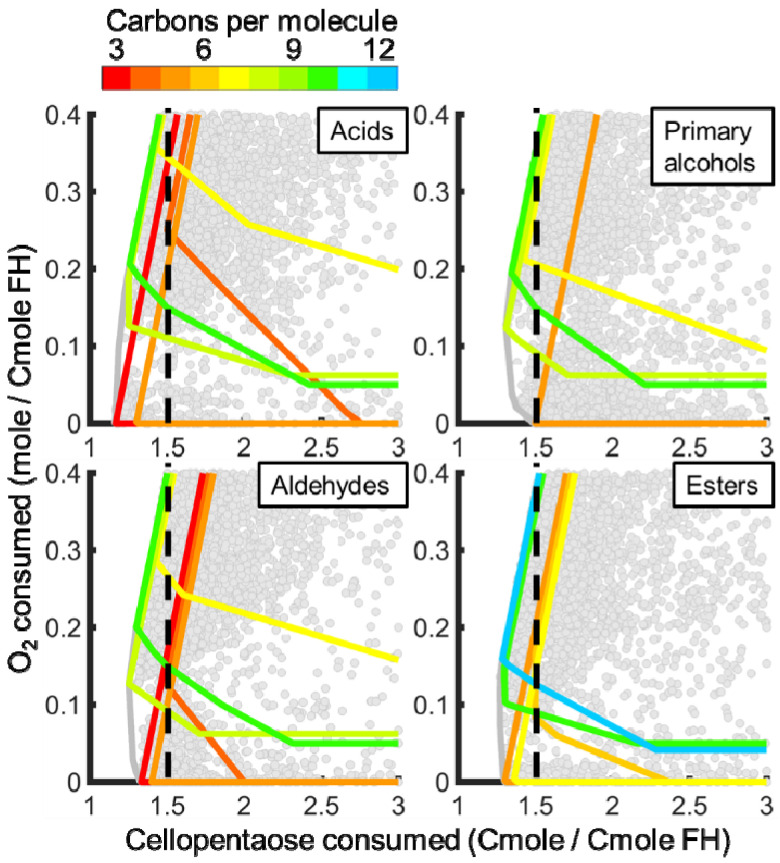
Ehrlich pathway derivative production strategies that minimize carbon and O_2_ requirements in *A. sarcoides* NRRL 50072 with respect to chain length and defining functional group. Metabolic strategies that minimize carbon source and O_2_ requirements to produce functionalized hydrocarbons (FHs) were identified (grey boundary, all possible metabolic strategies, elementary flux modes, are grey points). The colored lines denote metabolic strategies that produce FHs with a given number of carbons per molecule. Formate, acetate, acetaldehyde, and ethanol were not considered to be FHs for this study. Profiles for propanol, isobutyl alcohol, and isopentyl alcohol overlap. A reference carbon resource cost of 1.5 Cmole per Cmole of FH is denoted by the vertical dashed line to facilitate comparisons.

**Figure 5 jof-07-00435-f005:**
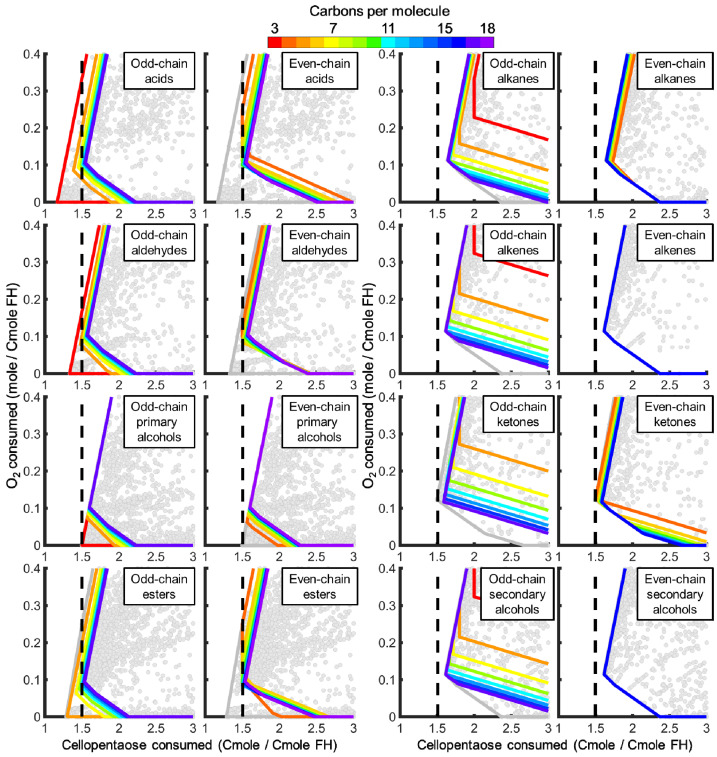
Fatty acid derivative production strategies that minimize carbon and O_2_ consumption in *A. sarcoides* NRRL 50072. Optimal O_2_ and carbon source utilization strategies that produce functionalized hydrocarbons (FHs) were identified to minimize resource costs (grey boundary, all possible metabolic strategies, elementary flux modes, and grey points). The colored lines denote metabolic strategies that produce FHs with a given number of carbons per molecule. Formate, acetate, acetaldehyde, and ethanol were not considered to be FHs for this study. Odd and even chain primary functional groups were initiated by propanoyl-CoA and acetyl-CoA, respectively, while the inverse is true for secondary functional groups. Resource profiles for even chain alkanes, even chain secondary alcohols appear as lines due to overlap. A reference carbon resource cost of 1.5 Cmole per Cmole of FH is denoted by the vertical dashed line to facilitate comparisons.

**Figure 6 jof-07-00435-f006:**
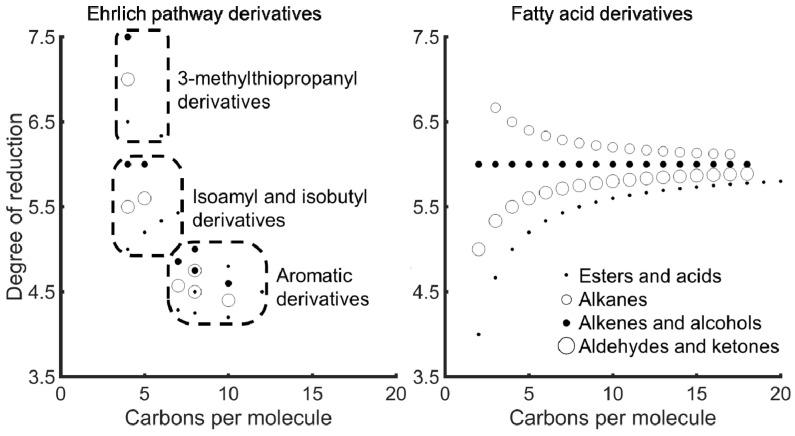
Degree of reduction of functionalized hydrocarbons plotted versus required carbons per molecule.

**Figure 7 jof-07-00435-f007:**
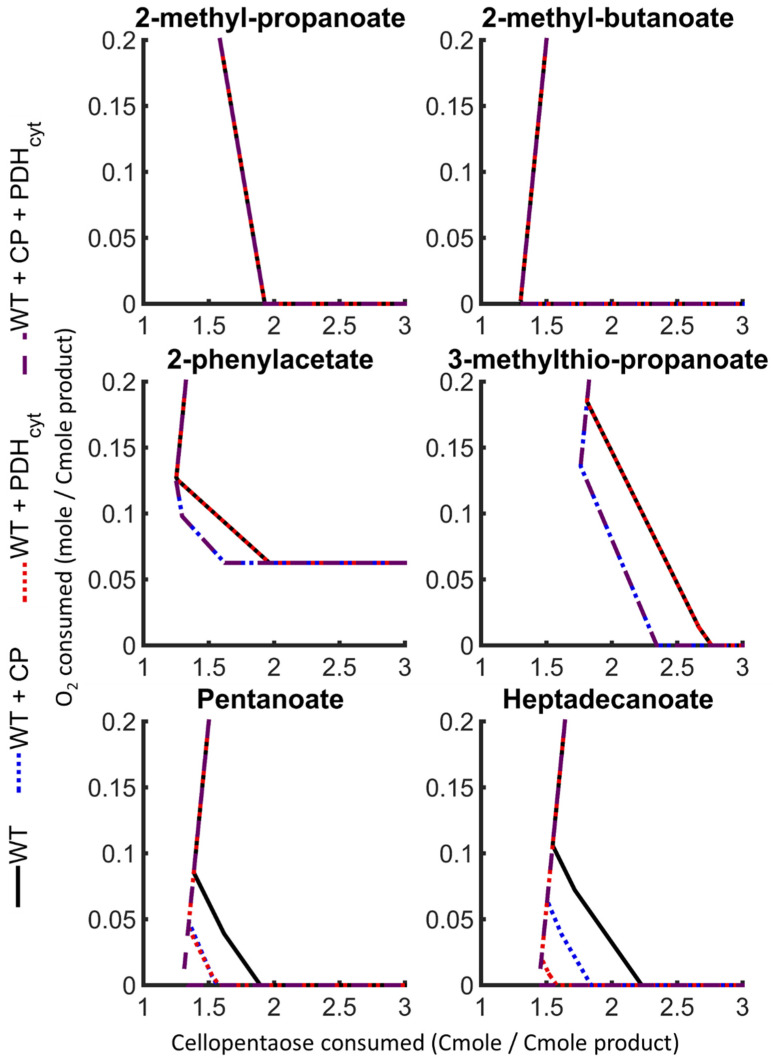
Theoretical effect of recombinant cellobiose phosphorylase (CP) and cytosolic pyruvate dehydrogenase (PDH_cyt_) activity on functionalized hydrocarbon production properties in *A. sarcoides* NRRL 50072 (black lines), with CP (blue dotted line), with PDH_cyt_ (red dotted line), and both (purple dashed line).

**Table 1 jof-07-00435-t001:** Commonly reported functionalized hydrocarbons from *A. sarcoides* NRRL 50072 ^a^ organized by hypothesized production pathway.

Fatty Acid Derivatives	Ehrlich Pathway Derivatives	Linoleic Acid Derivatives
1-butanol	2-methyl-1-propyl acid, alcohol, and esters	3-octanone
1-hexanol		1-octen-3-ol
Hexanoate	3-methyl-1-butyl acid, alcohol, and esters	
1-heptanol		
Octane	2-phenylethyl alcohol and esters	Miscellaneous
2-nonanone		Propyl-cyclopropane ^b^
C_2_, C_5_–C_10_ alkyl acetates	Phenyl methanol	2-pentyl-furan ^b^

^a^ All functionalized hydrocarbons were reported from at least three separate sources [1-3,5, and this study]; ^b^ Not included in the model.

**Table 2 jof-07-00435-t002:** Theoretical carbon source and O_2_ requirements to produce functionalized hydrocarbons based on wild-type *A. sarcoides* NRRL 50072 genomic potential and recombinant strains.

	Wild-type (WT)		WT + CP ^a^		WT + PDH_cyt_ ^a^		WT + CP + PDH_cyt_ ^a^	
Functional group ^b^	Carbon ^c^	O_2_ ^c^	Carbon ^c^	O_2_ ^c^	Carbon ^c^	O_2_ ^c^	Carbon ^c^	O_2_ ^c^
Acids	1.17 (1.2)	0.00	1.15 (1.2)	0.03	1.16 (1.2)	0.003	1.15 (1.2)	0.03
Aldehydes	1.25 (1.3)	0.13	1.23 (1.3)	0.08	1.25 (1.3)	0.13	1.23 (1.3)	0.08
Esters	1.29	0	1.29	0	1.29	0	1.29	0
Primary alcohols	1.31 (1.5)	0.12	1.28 (1.5)	0.09	1.31 (1.5)	0.12	1.28 (1.5)	0.09
Ketones	1.5 (2.7)	0.12	1.5 (2.2)	0.08	1.5 (2.0)	0.08	1.5 (1.7)	0.08
Alkenes	1.62 (2.4)	0.12	1.57 (2.0)	0.07	1.52 (1.7)	0.02	1.5	0
Secondary alcohols	1.62 (2.4)	0.12	1.57 (2.0)	0.07	1.52 (1.7)	0.02	1.5	0
Alkanes	1.64 (2.4)	0.11	1.60 (2.0)	0.07	1.54 (1.6)	0.01	1.53	0

^a^ Recombinant strains examined the effect of expressing cellobiose phosphorylase (CP) and a cytosolic pyruvate dehydrogenase complex (PDH_cyt_). ^b^ Each functional group was simulated in isolation. ^c^ Carbon and O_2_ are the Cmoles of cellopentaose and moles of O_2_, respectively, consumed per Cmole of functionalized hydrocarbons produced under carbon- or O_2_-limited conditions.

## Data Availability

All data are provided in the Appendix A.

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
