# Peer review of "In Silico Analysis of Functionalized Hydrocarbon Production Using Ehrlich Pathway and Fatty Acid Derivatives in an Endophytic Fungus"

_jof, 2021, doi:10.3390/jof7060435_

Round 1
Reviewer 1 Report
Dear Authors,
thank you for your interesting manuscript. I suggest to change the title from "Analysis of" to "In silico Analysis of"
In the summary and in the discussion you point out that your manuscript is "the first reported metabolic reconstruction of an endophytic filamentous fungus" please argument your statement taking into account the following paper:
Genomic sequencing, genome‑scale
metabolic network reconstruction, and in silico
flux analysis of the grape endophytic fungus
Alternaria sp. MG1
Yao Lu1, Chao Ye2, Jinxin Che3, Xiaoguang Xu1, Dongyan Shao1, Chunmei Jiang1, Yanlin Liu4 and Junling Shi1*
Author Response
RESPONSE: Thank you for the kind comments and suggestions. We appreciate the time you invested in helping us improve our document.
thank you for your interesting manuscript. I suggest to change the title from "Analysis of" to "In silico Analysis of"
RESPONSE: Thank you for the suggestion. We have changed the title to 'In silico analysis of functionalized hydrocarbon production using Ehrlich pathway and fatty acid derivatives in an endophytic fungus'
In the summary and in the discussion you point out that your manuscript is "the first reported metabolic reconstruction of an endophytic filamentous fungus" please argument your statement taking into account the following paper:
Genomic sequencing, genome‑scale
metabolic network reconstruction, and in silico
flux analysis of the grape endophytic fungus
Alternaria sp. MG1
Yao Lu1, Chao Ye2, Jinxin Che3, Xiaoguang Xu1, Dongyan Shao1, Chunmei Jiang1, Yanlin Liu4 and Junling Shi1*
RESPONSE: Thank you for bringing this interesting research to our attention. We have changed the wording in the abstract and introduction accordingly. Additionally, we have added a citation to this publication in the introduction.
Reviewer 2 Report
The work is of great interest and the experimental design is well structured.
I don't have any major comments to do about it, only minor:
- the quality of the figures should be improved. Sometimes the captions are hard to read and other times are cut out.
- Line 334-335 and following: leave a space between the end of the paragraph and the next paragraph's title.
Author Response
RESPONSE: Thank you for the supportive comments. They are appreciated.
The work is of great interest and the experimental design is well structured.
I don't have any major comments to do about it, only minor:
- the quality of the figures should be improved. Sometimes the captions are hard to read and other times are cut out.
RESPONSE: Thank you for the comment. We believe the resolution of some figures changed when they were compressed into the reviewer's file. We will watch for figure resolution and missing text as the file advances through the submission process.
- Line 334-335 and following: leave a space between the end of the paragraph and the next paragraph's title.
RESPONSE: Thank you for catching this. We have added spaces between the end of the paragraphs and the title of the next paragraph.